# Prevalence and Risk Factors for Hepatitis E Virus in Wild Boar and Red Deer in Portugal

**DOI:** 10.3390/microorganisms11102576

**Published:** 2023-10-16

**Authors:** Humberto Pires, Luís Cardoso, Ana Patrícia Lopes, Maria da Conceição Fontes, Sérgio Santos-Silva, Manuela Matos, Cristina Pintado, Luís Figueira, Ana Cristina Matos, João Rodrigo Mesquita, Ana Cláudia Coelho

**Affiliations:** 1Polytechnic Institute of Castelo Branco, 6001-909 Castelo Branco, Portugal; humberto.s.pires@gmail.com (H.P.); cpintado@ipcb.pt (C.P.); acmatos@ipcb.pt (A.C.M.); 2Animal and Veterinary Research Centre (CECAV), Department of Veterinary Sciences, University of Trás-os-Montes e Alto Douro (UTAD), 5000-801 Vila Real, Portugal; lcardoso@utad.pt (L.C.); aplopes@utad.pt (A.P.L.); mcfontes@utad.pt (M.d.C.F.); 3Associate Laboratory for Animal and Veterinary Sciences (AL4AnimalS), 5000-801 Vila Real, Portugal; 4School of Medicine and Biomedical Sciences (ICBAS), Porto University, 4050-313 Porto, Portugal; up202110051@edu.icbas.up.pt (S.S.-S.); jmesquita@outlook.com (J.R.M.); 5Centre for the Research and Technology of Agro-Environmental and Biological Sciences (CITAB), University of Trás-os-Montes e Alto Douro (UTAD), 5000-801 Vila Real, Portugal; mmatos@utad.pt; 6Research Center for Natural Resources, Environment and Society, Polytechnic Institute of Castelo Branco, 6001-909 Castelo Branco, Portugal; lmftfigueira@gmail.com; 7Quality of Life in the Rural World (Q-RURAL), Polytechnic Institute of Castelo Branco, 6001-909 Castelo Branco, Portugal; 8Epidemiology Research Unit (EPIUnit), Instituto de Saúde Pública da Universidade do Porto, 4050-600 Porto, Portugal; 9Laboratório Para a Investigação Integrativa e Translacional em Saúde Populacional (ITR), 4050-600 Porto, Portugal

**Keywords:** ELISA, hepatitis E virus, Portugal, red deer, risk factors, wild boar

## Abstract

Hepatitis E virus (HEV) is a zoonotic foodborne virus with an annual infection prevalence of 20 million human cases, which seriously affects public health and economic development in both developed and developing countries. To better understand the epidemiology of HEV in Central Portugal, a cross-sectional study was conducted from 2016 to 2023 with sera samples from wild ungulates. The seroprevalence and risk factors for HEV seropositivity were evaluated in the present study. Specifically, antibodies against HEV were determined by a commercial enzyme-linked immune-sorbent assay (ELISA). Our results show that in the 650 sera samples collected from 298 wild red deer and 352 wild boars in Portugal, 9.1% red deer and 1.7% wild boar were positive for antibodies to HEV. Regarding age, the seropositivity in juvenile wild ungulates was 1.3%, whereas it was 7.2% in adults. Logistic regression models investigated risk factors for seropositivity. The odds of being seropositive was 3.6 times higher in adults than in juveniles, and the risk was 4.2 times higher in red deer than in wild boar. Both wild ungulate species were exposed to HEV. The higher seroprevalence in red deer suggests that this species may make a major contribution to the ecology of HEV in Central Portugal. Further research is needed to understand how wildlife affects the epidemiology of HEV infections in Portugal.

## 1. Introduction

The most common cause of acute, enterically transmitted hepatitis in developing countries is the hepatitis E virus (HEV) [1]. HEV is a foodborne zoonotic virus with a worldwide infection prevalence of 20 million human cases, causing severe implications for public health and economic progress in developed and developing nations [2,3]. In Europe, more than 20,000 confirmed hepatitis E cases were reported between 2005 and 2015 [4]. Besides humans, HEV infects various mammalian hosts [5]. HEV belongs to the family Hepeviridae, subdivided into the subfamilies Parahepevirinae and Orthohepevirinae. Members of the Parahepevirinae subfamily infect salmon and trout. In contrast, mammals and birds are infected by members of the Orthohepevirinae subfamily, which can be further divided into four genera: *Paslahepevirus*, *Avihepevirus*, *Rocahepevirus*, and *Chirohepevirus* [6]. 

HEV is a small virus quasi-enveloped with an icosahedral capsid enclosing the viral genome [7]. The positive-sense, single-stranded RNA comprising the genome is 6.4–7.3 kb long and has four partially overlapping open reading frames [8,9].

Today, HEV is referred to as *Paslahepevirus balayani*. Eight genotypes of HEV are known, namely HEV-1 through HEV-8, with genotypes 1 and 2 exclusively infecting humans, genotypes 3, 4, and 7 infecting both humans and animals, and genotypes 5, 6, and 8 only infecting animals [10]. HEV-1 and HEV-2 are prevalent in developing countries and can cause significant gastrointestinal symptoms in humans. These symptoms are often attributed to consuming contaminated water and food [11,12]. Zoonotic genotypes 3 and 4 are primarily transmitted through consumption of contaminated pork and meat products, or contact with infected animals, particularly pigs (*Sus scrofa domesticus*) [13]. In industrialized countries, HEV-3 and HEV-4 typically cause sporadic and pauci-symptomatic illnesses. In humans, HEV typically manifests with mild clinical characteristics involving increased liver enzyme levels, jaundice, and non-specific symptoms like loss of appetite, fatigue, and abdominal discomfort. These clinical signs often closely resemble symptoms observed in various other liver disorders, and they usually persist for a duration of 1 to 6 weeks. Though most cases resolve independently, chronic hepatitis E infections have been documented in individuals with weakened immune systems, especially among organ transplant recipients using immunosuppressive drugs [14,15].

After being identified in domestic pigs in the United States in 1997, swine HEV strains have been recognized globally in both domestic and wild pig populations, showing considerable variations in their prevalence [16,17]. HEV-3 and HEV-4 genotypes have been detected in humans, as well as in *Sus scrofa domesticus* (domestic pigs) and *Sus scrofa* (wild boars) [12]. The HEV-3 and HEV-4 genotypes, found in various animal species such as wild ungulates, exhibit significant similarities with the genotypes found in humans. These similarities support the zoonotic nature of these two genotypes [17,18,19]. Genotype 4 is common to China and is the predominant genotype there [20]. In contrast, genotype 3 is common to all other parts of the world and is the most common HEV genotype in Europe [21,22,23,24]. HEV genotypes 5 and 6 have only been identified in Japanese wild boar (*Sus scrofa leucomystax*) [22], whereas HEV genotypes 7 and 8 have recently been detected in camels in China (*Camelus bactrianus*) and the Middle East (*Camelus dromedarius*) [25,26].

The unequivocal link between the consumption of infected wild ungulate meat and human infection has been previously described [27]. This includes the finding that consuming contaminated wild boar meat can result in human infection. [28]. Cases of HEV have been directly connected to the consumption of raw deer meat by the presence of identical HEV strains in the consumed deer meat and the respective patients [29]. In immunocompetent human subjects, HEV causes sporadic self-limiting infections with low mortality. Conversely, in immunocompromised patients, HEV can induce several dangerous clinical symptoms [30,31] and it may also cause neurological manifestations [32]. However, mortality can reach higher levels in pregnant women (20%), as they can develop severe hepatitis [33,34,35]. The main HEV reservoirs are domestic pigs and wild boar [36,37]. However, zoonotic strains have also been identified in deer, rabbits, chickens, mongooses, rats, ferrets, fish, and camels. This range of hosts continues to expand over time, showcasing instances of infections crossing between different species [38,39]. The development of HEV antibodies in pigs occurs after the natural decrease in maternal antibody levels, typically around 8–10 weeks of age. Initially, IgM anti-HEV antibodies reach their highest levels alongside a peak of viral shedding through feces, followed by IgG anti-HEV antibodies peaking when the virus is being eliminated through fecal matter. The liver is the targeted organ from which HEV spreads to various tissues and organs through hematic diffusion [40]. A previous study suggests that deer species may be the natural reservoir for HEV-3 [41], but this evidence needs further investigation to determine which situations deer serve as a reservoir.

Although HEV infections have been detected in wild ungulates in many countries around the world [42,43,44,45,46], there has only been one seroepidemiological study on HEV in wild ungulates in Portugal, and this focused only on wild boar. The epidemiological situation of wild animals, such as deer, in Portugal is practically unknown; however, this knowledge is fundamental for the application of prevention and control measures. As such, this study aimed to assess the seroprevalence and risk factors associated with HEV in wild boar and red deer in Central Portugal.

## 2. Materials and Methods

From 2016 to 2023, a survey was conducted on serum samples randomly collected from free-ranging wild ungulates hunted and legally killed by hunters in east-central Portugal to investigate the presence of HEV. The first hunted animals were sampled up to a total of 10 times per year on each side and this sampling process was repeated each year in new hunted animals.

Sampled municipalities included Alcafozes (*n* = 16), Castelo Branco (*n* = 30), Cegonhas (*n* = 8), Crato (*n* = 41), Fratel (*n* = 35), Granja (*n* = 10), Idanha-a-Nova (*n* = 29), Lousa (*n* = 44), Marvão (*n* = 31), Mata (*n* = 40), Monforte (*n* = 10), Monte Fidalgo (*n* = 76), Niza (*n* = 26), Ponte de Sor (*n* = 25), Portalegre (*n* = 49), Rosmaninhal (*n* = 39), Sarnadas do Ródão (*n* = 40), Tostão (*n* = 9), Vila Velha de Ródão (*n* = 64), Vale de Figueira (*n* = 6), and Vale Pousadas (*n* = 22).

These regions are home to the largest population of wild ungulates in Portugal. A veterinarian conducted a comprehensive examination of a total of 650 wild ungulates, representing two distinct species: 352 wild boar (*S. scrofa*) and 298 red deer (*Cervus elaphus*). Available data on age, sex, body condition, and capture location were utilized to provide insights into the distribution of seropositive individuals. The wild ungulates were divided into two groups according to age: juveniles and adults. Wild boars were considered juveniles until they were 8 months old, at which point they can become pregnant. However, red deer were only considered adults when they were over 1.5 years old. Clinical signs were considered whenever the animal presented with one or more of the following signs: poor body or coat condition, or macroscopic lesions of any kind in external or internal organs.

Blood samples were obtained from the heart or thoracic cavity of the animals during the hunting season, which was between October and February each year. Blood was allowed to clot at environmental temperature and was transported afterward to the laboratory. Blood samples were then centrifuged at 1500× *g* for 10 min and the separated serum samples were kept at −20 °C until further testing. All sera were checked for the presence of antibodies to HEV in wild ungulates using a commercial ELISA kit (ID Screen^®^ Hepatitis E Indirect Multi-species ID.vet, Montpellier, France), under the manufacturer’s recommendations and following their guidelines for the interpretation of results. This is a duplicate-well test in which even-numbered wells are coated with a recombinant antigen from the capsid of HEV genotype 3 expressed in *Baculovirus*, and odd-numbered wells are uncoated. This ELISA employs a combination of protein A/G and horseradish peroxidase to identify IgG antibodies. Optical densities (OD) of the tested samples and positive and negative controls were measured by an ELISA plate reader at 450 nm. 

The OD ratio of the sample and positive control (S/P) was calculated for each sample, as follows, with a cut-off value set at 70 (S/P%):

To calculate the net OD results: net OD = OD_even well_ − OD_odd well_.

The test was validated if the net mean value of the Positive Control OD (OD_pc_) was greater than 0.350: net OD_pc_ > 0.350. The ratio of the net mean values of the Positive and Negative Control ODs (OD_pc_ and OD_nc_) was greater than 3. In the formula, the absolute value of the net OD_nc_ was used: net OD_pc_/|net Od_nc_| > 3.

For each sample, the S/P percentage (S/P%) was calculated:$$S/P=\frac{{net\;OD}_{sample}}{{net\;OD}_{pc}}\times 100$$

Following the manufacturer’s instructions, readings equal to or below 60% were considered negative, readings equal to or greater than 70% were considered positive, and readings between 60 and 70% were considered doubtful. Doubtful results were run in duplicate wells, using the same protocol. According to the manufacturer’s data and preliminary studies [47,48,49], the ELISA used had a specificity of 100%.

### Statistical Analysis

The outcome variable was dichotomized as positive versus not positive to identify any risk factors associated with seropositivity. A Chi-square test was used to assess significant differences among the groups. Multiple logistic regression was used to model the odds ratio (OR) and its 95% confidence interval (CI) of being seropositive related to the variables. Significant potential risk factors at *p* < 0.05 (two-tailed; alpha = 0.05) were then evaluated using stepwise regression to construct a multiple model (Wald test stepwise p-Wald value to enter *p* < 0.05). The multiple logistic model was developed using a stepwise approach. Backward elimination followed by a forward selection for each variable at a time was performed using a likelihood ratio test at each step with 0.05 (two-tailed; alpha = 0.05) as the significance level for removal or entry. The fit of the models was assessed using the Hosmer and Lemeshow goodness-of-fit test [50]. The model was rerun until all remaining variables presented statistically significant values (*p* < 0.05). All statistical analyses were performed using SPSS^®^ 25.0 software for Windows.

## 3. Results

The wild ungulates included in this study comprised 352 (54.2%) wild boar and 298 (45.8%) red deer (Figure 1). 

By age group, adults represented 64.2% (n = 417) of the total number of animals, whereas juveniles made up 35.8% (n = 233). Following the European Regulations (EU Reg.), each animal received ante- and post-mortem examinations, respectively performed by hunters (in agreement with the EU Reg. No. 853/2004) and by veterinary sanitary authorities (EU Reg. No. 625/2017).

In the present study, the overall seroprevalence was 5.1% (n = 33; 95% CI: 3.5–7.1%). 

Among the species that tested positive, red deer had a significantly higher prevalence (9.1%; 95% CI: 6.1–12.9%) than wild boar (1.7%; 95% CI: 0.6−3.7%; *p* < 0.001).

Serologic reactivity data according to species, sex, age, and clinical signs are presented in Table 1. The seroprevalence values among male and female animals were 6.6% (95% CI: 4.2–9.7%) and 3.3% (95% CI: 1.6–6.0%), respectively (*p* = 0.054) (Table 1). Regarding age, the lowest value of seroprevalence (1.3%; 95% CI: 0.3–3.7%) was found in juveniles and the highest (7.2%; 95% CI: 4.9–10.1%) in adults. Statistically significant differences were observed between these groups (*p* < 0.001). Furthermore, there was a significant difference in seropositivity results related to the presence (1.7%; CI: 0.2–6.0%) or absence of clinical signs (5.8%; 95% CI: 4.0–8.2%) in the studied species (*p* = 0.037) (Table 1).

Regarding municipal distribution, anti-HEV antibodies were found in nine municipalities: three red deer from Castelo Branco (10.0%; 3/30 wild ungulates), four red deer from Crato (9.8%; 4/41 wild ungulates), one red deer from Fratel (2.9%; 1/35 wild ungulates), two wild boar from Lousa (4.5%; 2/44 wild ungulates), four red deer from Marvão (12.9%; 4/31 wild ungulates), two wild boar from Niza (7.7%; 2/26 wild ungulates), four red deer and two wild boar from Rosmaninhal (15.4%; 6/39 wild ungulates), eight red deer from Vila Velha de Ródão (12.5%; 8/64 wild ungulates), and three red deer from Vale Pousadas (13.6%; 3/22 wild ungulates).

Three variables were associated (*p* < 0.05) with seropositivity in wild ungulates. Seropositivity significantly correlated with the following factors (Table 1): age, species, and clinical signs. These variables were included in the multiple model. A backward stepwise conditional logistic regression was employed using all the statistically significant variables above. The multiple logistic regression analysis of the OR for being seropositive to potential risk factors is presented in Table 2. At the individual level, the odds of HEV seropositivity were found to be higher for adult animals, OR = 3.66 (95% CI: 1.72–18.11%), and for red deer, OR = 4.2 (95% CI: 1.64–10.69%). 

## 4. Discussion

Each year, foodborne pathogens, including the HEV, result in numerous infections across various continents. Ingestion of contaminated food of animal origin exposes consumers to a risk of infection. HEV has been identified as an emergent public health risk in several European countries [51].

Annually, HEV leads to a considerable number of infections through zoonotic transmission. Moreover, it has gained recognition as an occupational disease due to the prevalence of antibodies found in the blood samples of professionals like veterinary workers, slaughterhouse employees, hunters, and similar occupations [12]. Individuals such as veterinary food inspectors, pig breeders, hunters, pork-product sellers, and meat-processing workers are frequently exposed to the risk of HEV infection [52]. Even in developed countries, where the origins of human hepatitis E cases were previously thought to be rare, detection of HEV in sewage, water sources, coastal and surface waterways, drinking water, and produce raises environmental safety concerns [38,53].

Epidemiological studies utilize molecular biology techniques like real-time PCR assays and with seroprevalence screening methods involving the detection of immunoglobulin G (IgG) and/or immunoglobulin M (IgM) as highly reliable and established approaches for diagnosing and assessing the epidemiological situation [54].

This cross-sectional study shows that approximately 5.1% of surveyed wild game that were hunted for human consumption had antibodies for HEV. As far as we know, the present study is the first serological study conducted on red deer in Portugal, and the largest seroprevalence study on wild boar in Portugal to date.

Human HEV infections caused close to 21,000 clinical cases in Europe between 2005 and 2015. Most were limited to specific countries (e.g., Germany, France, Italy, and Spain [4]). Serological studies have shown that people are in danger of infection, as reported by Raji et al. [55], who found substantial seroprevalence titres in hunters and in blood donors [31]. Consumption of infected (raw or undercooked) pork liver or wild boar meat products is the main mode of HEV infection in humans, and traditional home-cooked foodstuffs, in particular, allow HEV persistence in specific geographical locations [18,31,56,57]. Occupational exposure of workers to swine and sheep represents another risk factor associated with seropositivity [58,59]. A study involving 114 individuals involved in agricultural activity was conducted in Portugal. The results revealed that 30.7% of workers had antibodies to HEV, whereas the general population had a seropositivity of 19.9% [58].

In the Portuguese human and animal population, there have been reports of evidence of infection detected by molecular and serological methods [60,61,62,63,64,65,66]. In Portugal, HEV has previously been detected using molecular methods in wild ungulates. Namely, HEV was detected in two red deer with a prevalence of 2.1% [60], 20 wild boar livers (25.0%), and four wild boar stools in two different studies (10 and 2.8%) [61,62]. 

In the Iberian Peninsula, serological reports of HEV in wild ungulates have ranged between 8.2–57.4% in wild boar [11,42,67,68,69,70,71] and 4–13.9% in red deer [68,69,70,71,72,73]. The seroprevalence in wild ungulates found in the present study is lower than what was found in previous studies from the Iberian Peninsula. Animals with low serum viral titres, diagnosed as negative for HEV, can still transmit the disease, making it necessary to evaluate and clarify HEV diagnostic protocols in animals intended for human consumption [58].

Moreover, the values found in this serological study are lower than those found in other similar European studies. In Italy, a study found seroprevalences between 87.3 and 100% in pigs [74], and similar results have been found in white-tailed deer (*Odocoileus virginianus*) in Finland (1.4%) [75]. Interestingly, other studies in Italy also detected HEV RNA in 14.3–83.7% of liver samples [11,43,44,45,76,77,78], and in farmed ruminants, the majority of HEV genotypes discovered to date are from the zoonotic HEV3 and HEV4 [79].

Although the seroprevalence found in this study appears to be low, the etiological agent seems to infect the wild ungulates under evaluation. Seropositive variance between red deer and wild boars indicates the presence of current infections and previous exposures in the examined animals.

In the statistical analysis, only two factors were associated in the final model: age and species. The high seropositivity level reported in adult wild ungulates supports prior research. It can be explained by longer-term exposure to HEV in the environment, with similar studies on wild ungulates from other countries in Europe finding the same pattern [80,81]. Furthermore, this tendency has been observed in humans as well, with anti-HEV seroprevalence increasing with age [82,83] and seropositivity associated with the female gender, probably due to the potential role of sexual hormone peaks, which may increase host receptivity to infection [84,85]. In the present study, the test chosen to detect HEV infection proved to be practical, rapid, and cheaper for detecting previous HEV exposure.

Domestic pigs play a prominent role as a reservoir for the HEV among wildlife and domestic animal species. Surveillance for HEV is crucial globally to address knowledge gaps related to its transmission and reservoirs, particularly considering its zoonotic potential [86]. From an epidemiological standpoint, domestic pigs are closely linked with other important wild species, such as *S. scrofa* (wild boars) and wild ruminants, which serve as additional sources for the environmental spread of the virus [18,28,87,88,89]. The results of this study should be interpreted from a One Health perspective. The interaction between different species, particularly wild ungulates and domestic ruminants in their shared habitats, can facilitate the transmission of infectious agents such as HEV. Previous studies reveal a similar HEV RNA detection rate in red deer and wild boars inhabiting the same regions, implying that the same HEV-3 variant is frequently transmitted among various ungulate species [70,90] and ungulate species like deer serve as a true host for HEV. However, wild and domestic pigs are still the principal cause of infection for ruminants living in the same regions [90]. Additionally, the proximity of small ruminants and wild animals to humans increases the risk of pathogen transmission to people [91]. 

However, it is important to interpret these findings cautiously, as the present study has some limitations. Specifically, this study involved a cross-sectional design with self-selection of animals. The authors alert to the possibility that the time of sampling may have affected the results of HEV seroprevalence, as elevated ambient temperature is known to influence immunoglobulin G and immunoglobulin G subclasses [92]. A better understanding of HEV epidemiology and infection risks is not possible without determining which animals are viremic and when transmission occurs.

The results of this ELISA-based survey indicate that HEV infection is widely distributed among red deer and wild boar from the central region of Portugal. Considering the lack of similar studies in the country, particularly on red deer, our results could contribute to the effective control of HEV in wildlife. These results also confirm that the seroprevalence of HEV infection in red deer and wild boar have been underestimated in Portugal.

In this study, we observed a higher seroprevalence of HEV in deer compared with wild boar, despite the conventional understanding that wild boar are often considered the primary reservoir of HEV in the environment. Several factors could have contributed to this apparent discrepancy: Multiple genotypes and strains of HEV are known to circulate, and different strains may have varying host preferences. Also, though wild boar is often considered the primary reservoir in the wild, cross-species transmission can occur from other species. Deer and wild boar may share habitats and water sources, thus facilitating virus transmission between the two species. These findings warrant further investigation into the underlying factors driving the observed seroprevalence patterns in deer and wild boar.

The present study aims to contribute to the detection and identification of potential future threats related to HEV infections in wild ungulates. Wildlife monitoring plays a crucial role as it enables the identification of changes in disease occurrence among wild populations [93,94,95]. Regularly monitoring and studying the health status of wild ungulates can detect shifts or patterns in transmission dynamics in the wildlife population [95]. This information is essential for understanding the potential risks and impact associated with HEV infections.

## 5. Conclusions

In conclusion, the present study emphasizes the importance of a One Health, multidisciplinary approach for assessing wild boar and red deer exposure to HEV in Central Portugal and for controlling HEV disease. Further research on the role of wildlife in the epidemiology of HEV infection should be conducted. By addressing concerns associated with wildlife reservoirs and integrating them into disease control, better efforts can be made to safeguard animal and human health in the face of HEV.

## Figures and Tables

**Figure 1 microorganisms-11-02576-f001:**
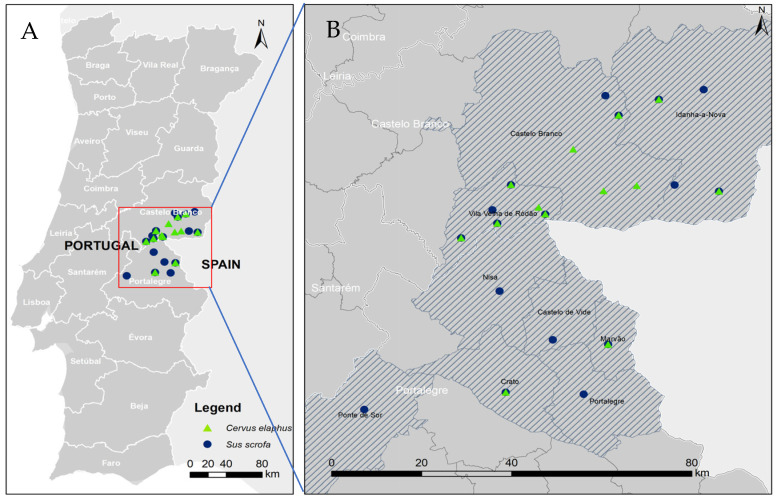
Localization of wild ungulate sample collection for seroepidemiological study in Portugal. Image (**B**) represents a magnification of the red box in image (**A**).

**Table 1 microorganisms-11-02576-t001:** Screening for anti-HEV antibodies in free-ranging wild ungulates from Central Portugal.

	Wild Boar n = 352 Positive/Total (%)	CI 95%	Red Deer n = 298 Positive/Total (%)	CI 95%	No. Anti-HEV n = 650 Positive/Total (%)	CI 95%
Sex	*p* = 0.524		*p* = 0.059		*p* = 0.054	
Male	4/190 (2.1%)	0.6–5.3%	19/159 (11.9%)	7.4–18.0%	23/349 (6.6%)	4.2–9.7%
Female	2/162 (1.2%)	0.015–4.4%	8/139 (5.8%)	2.5–11.0%	10/301 (3.3%)	1.6–6.0%
Age	*p* = 0.390		*p* = 0.016 *		*p* < 0.001 *	
Juvenile	2/178 (1.1%)	0.14–4.0%	1/55 (1.8%)	0.0–9.7%	3/233 (1.3%)	0.3–3.7%
Adult	4/174 (2.3%)	0.6–5.8%	26/243 (10.7%)	7.1–15.3%	30/417 (7.2%)	4.9–10.1%
Clinical Signs	*p* = 0.876		*p* = 0.286		*p* = 0.037 *	
Absence	4/245 (1.6%)	0.5–4.1%	27/287 (9.4%)	6.3–13.4%	31/532 (5.8%)	4.0–8.2%
Presence	2/107 (1.9%)	0.2–6.6%	0/11 (0.0)	0.0–2.9%	2/118 (1.7%)	0.2–6.0%
Total	6/352 (1.7%)	0.6–3.7%	27/298 (9.1%)	6.1–12.9%	33/650 (5.1%)	3.5–7.1%

* *p* < 0.05; CI: confidence interval.

**Table 2 microorganisms-11-02576-t002:** Risk factors associated with HEV infection of wild ungulates in the Centre of Portugal.

Risk Factor	β ^a^	S.E. β ^b^	*P*	OR ^c^	95% CI ^d^ (OR)
Age	1.298	0.641	0.043		
Juvenile				1	
Adult				3.662	1.042–12.867
Species	1.431	0.479	0.003		
Wild boar				1	
Red deer				4.184	1.637–10.692

^a^ β: logistic regression coefficient; ^b^ S.E. β: standard error; ^c^ OR: odds ratio; ^d^ CI: confidence interval.

## Data Availability

The data presented in this study are available on request from the corresponding author.

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
