# Peer review of "Prevalence and Risk Factors for Hepatitis E Virus in Wild Boar and Red Deer in Portugal"

_microorganisms, 2023, doi:10.3390/microorganisms11102576_

Round 1

Reviewer 1 Report

Line 31. Does the ELISA have a validation from OMSA?

Line 60. You just stated that HEV1 and HEV2 only infect humans and later, that HEV1 and HEV2 are prevalent in developing countries, and can cause significant gastrointestinal symptoms/signs in humans and animals. This is contradictory, please clarify.

Line 66. I think it will be better if you add the scientific name of the pigs here.

Line 85-86.  Add: (Sus scrofa leucomystax) and (Camelus dromedarius and C. bactrianus) according with the references you provided.

Line 98. Please add adequate references.

Line 101-103. This implies that they are exposed to the virus routinely, from very early on in their development. What is the overall frequency of HEV in domestic pigs?

Line 110. At this point it is very clear why you are looking for HEV antibodies in the wild, but nothing about red deer, The result from one study suggests that deer species can be the natural reservoir for HEV3, can you add a comment about this?

Line 112. Why you did not take a liver biopsy from the animals that were killed? So you have a record of animals that were captured and recaptured more than once?

Line 115. Does this mean that you can have an idea of the lifespan of the HEV antibodies in both species?

Line 116-120. Please add (as supplementary material, it is the case, a map showing the localities)

Line 124. You can use it here:  C. elaphus.

Line 173-180. This can be presented either in a Table or Figure.

Line. 210-213. Please clarify if you made capture and recapture follow-up, and present the dynamic of the antibodies in these animals.

Line 223-227. Why you didn´t try to isolate in culture and/or detect the HEV by molecular biology? You could have access to the liver of the killed animals.

Author Response

REVIEWER 1

Q1. Line 31. Does the ELISA have a validation from OMSA?

A1. This ELISA kit does not have a validation from OMSA. In fact, and to the best of our knowledge, no ELISA for the screening of HEV antibodies has validation from OMSA. It has been, however, often used for the screening of HEV antibodies in wild boar and deer in the past (see examples below):

https://www.ncbi.nlm.nih.gov/pmc/articles/PMC7600272/

https://www.ncbi.nlm.nih.gov/pmc/articles/PMC5176221/

Q2. Line 60. You just stated that HEV1 and HEV2 only infect humans and later, that HEV1 and HEV2 are prevalent in developing countries, and can cause significant gastrointestinal symptoms/signs in humans and animals. This is contradictory, please clarify.

A2. The reviewer is right in her/his remark. HEV-1 and HEV-2 only affect humans and, thus, only cause significant gastrointestinal symptoms/signs in humans. We have rephrased text for clarity.

Q3. Line 66. I think it will be better if you add the scientific name of the pigs here.

A3. Thanks for the comment. The scientific name has been added.

Q4. Line 85-86. Add: (Sus scrofa leucomystax) and (Camelus dromedarius and C. bactrianus) according with the references you provided.

A4. Thanks for the comment. The scientific names have been added.

Q5. Line 98. Please add adequate references.

A5. We suggest adding the following references to the text to support the sentence

https://pubmed.ncbi.nlm.nih.gov/17606958/

https://pubmed.ncbi.nlm.nih.gov/32522266/

Q6. Line 101-103. This implies that they are exposed to the virus routinely, from very early on in their development. What is the overall frequency of HEV in domestic pigs?

A6. the reviewer is correct, indeed pigs are exposed to the virus routinely, from very early age. According to a recent systematic review, seroprevalence and RNA positivity in the 0-4 month-old group was found to be 42.19% and 17.62%, showing that infection occurs early in life.  Globally, in pigs anti-HEV seroprevalence was of 59.33% and  HEV RNA positivity was 12.71%.

https://www.ncbi.nlm.nih.gov/pmc/articles/PMC8686068/

Q7. Line 110. At this point it is very clear why you are looking for HEV antibodies in the wild, but nothing about red deer, The result from one study suggests that deer species can be the natural reservoir for HEV3, can you add a comment about this?

A7. Thanks for the comment. We have inserted a comment about this and added a reference.

https://pubmed.ncbi.nlm.nih.gov/26518619/

Q8. Line 112. Why you did not take a liver biopsy from the animals that were killed? So you have a record of animals that were captured and recaptured more than once?

A8. Thanks for your comments. We will take the suggestions for next studies. The aim of this study was only to assess seroprevalence and risk factors associated with HEV. Taking a liver sample would have implied a different logistic and budget, and in that time frame we are not sure whether would have found antibodies in deer. We will take that suggestion for the future.

Q9. Line 115. Does this mean that you can have an idea of the lifespan of the HEV antibodies in both species?

A9. Thanks for your comments. No, we only have the idea if the occurrence is similar or diferente because they are new hunted animals, each year.

Q10. Line 116-120. Please add (as supplementary material, it is the case, a map showing the localities)

A10. Thanks for your comments. We added a map showing the localities.

Q11. Line 124. You can use it here: C. elaphus.

A11. The correction has been made.

Q12. Line 173-180. This can be presented either in a Table or Figure.

A12. Thanks for the comments. We produce a Figure, but it was difficult to read. We also produced a table, which is easier to read. We have opted for not using, if that is not mandatory.

Q13. Line. 210-213. Please clarify if you made capture and recapture follow-up, and present the dynamic of the antibodies in these animals.

A13. Thanks for the comments. This was a cross-sectional study without follow-up. We have reformulated the sentence.

Q14. Line 223-227. Why you didn´t try to isolate in culture and/or detect the HEV by molecular biology? You could have access to the liver of the killed animals.

A14. Thanks for your comments. We will take the suggestions for next studies. The aim of this study was only to assess the seroprevalence and risk factors associated with HEV. Taking a liver sample implies a different logistic and budget, and in that time we are not sure that we will find antibodies in deer. We will take that suggestion for the future.

Author Response

REVIEWER 2

Q1. Pires and colleagues present: Prevalence and Risk Factors for Hepatitis E Virus in Wild Boar and Red Deer in Portugal.

A1. Thanks for the revision performed.

Q2. Title: There is no need to have the Taxonomical genus and species in the title of your manuscript. Put them in the introduction. Also, you do not need to have the HEV in parentheses because you also do that in the abstract and in the introduction. Define it in one place and refer to it as HEV thereafter.

A2. Thanks for the comments. The sentence has been revised according the Reviewer’s suggestion.

Q3. Abstract

Do not put your confidence intervals in the abstract. Keep them in the results and possibly the discussion.

A3. Thanks for the comments. The sentence has been revised according to the Reviewer’s suggestion.

Q4. Introduction:

You begin talking about the genotypes of HEV and refer to them as HEV-1 through HEV-8. This is the correct designation. Later you begin referring to them as HEV1 and HEV2, which should be HEV-1 and HEV-2. Please be consistent throughout the manuscript when you refer to the genotypes.

A4. Thanks for the corrections. The sentence has been revised according to the Reviewer’s suggestion.

Q5. Lines 74-77: The genotypes designated as swine HEV should be identified in this paragraph not the next one. 74-86 should be one paragraph.

A5. Thanks for the comments. The sentence has been revised according to the Reviewer’s suggestion.

Q6. Line 87-89: you should not start a sentence with Including. It does not read well. Please reword the two sentences.

A6. Thanks for the comments. The sentence has been reworded.

Q7. Methods:

This description of serum collection inadequate. How were you able to collect serum samples from ungulates randomly killed by hunters? Were the samples heat inactivated before testing? You need a brief description of the commercial plate and how it was used. Were the samples tested in duplicate or triplicate?

A7. Thanks for the comments. The methodology has been revised according to the Reviewer’s suggestion, and information was added to the manuscript. The samples were tested in duplicate.

Results:

Q8. Line 153: Sentence should read “The population of wild ungulates studied comprised…”

A8. Thanks for the corrections. The sentence has been revised as per the Reviewer’s suggestion.

Q9. Line 158: “The global seroprevalence of HEV infection was…” How does this data fit in with your study? No time frame or species described either. Where did you get this data on global prevalence?

A9. The Reviewer is correct, we have rephrased text to read as “In the present study, the overall seroprevalence was…”

Q10. Line 167: This sentence is confusing. Presence or absence of what? Please reword for better understanding of what you are trying to say.

A10. Thanks for the comments. The sentence has been revised according to the Reviewer’s suggestion.

Discussion:

Q11. Line 214: Do not use the word “around” when stating a number. Approximate, over or under are better descriptors.

A11. Thanks for the comments. The sentence has been revised according to the Reviewer’s suggestion.

Q12. Line 254: Reword this sentence. “over the world” should be reworded.

A12. Thanks for the comments. The sentence has been revised according to the Reviewer’s suggestion.

Figures and Tables:

Q13. Table 1 should be redone. It is confusing. There are no explanations at the bottom describing what those p values are and how you got them. It would be better to have more columns keeping the percentages and CI separate. There is no description of what the clinical signs were.

A13. Thanks for the comments. Table 1 has been redone. Information has been added.

Q14. Juvenile is spelled incorrectly. Also define what you consider Juvenile for each species. Is it legal to hunt juveniles in Portugal?

A14. Thanks for the comments. The correction has been made and information about what we consider juvenile for each species has been added. In Portugal it is legal to hunt juveniles in the age of the animals of the manuscript.

Q15. Overall, this is an interesting study, but I believe that because HEV is viremic that RT-PCR should have been performed in addition to ELISA. I found that the methods were inadequately described, and the tables are confusing. I would consider approving this manuscript were it to be revised.

A15. We agree with the Reviewer and screening for HEV RNA would have provided a clearer picture regarding the presence of active infection in these animals. However, during the pilot phase of this study, we have seen difficulties in maintaining steady refrigeration of samples from the sampling sites to the laboratory. Blood samples had to be obtained from the heart or thoracic cavity of the animals after hunting of the animal and this blood had to be allowed to clot at environmental temperature to separate the serum, potentially hampering future RNA detection.

In this sense, we have opted for expanding the sample size to the number of 650 wild ungulates and opted to focus on HEV antibody detection, hence providing more robust meaning from the risk analysis approach.

Reviewer 3 Report

Pires H. and coworkers investigated the epidemiology of hepatitis E virus in the Centre of Portugal in sera samples collected from wild ungulates (wild boars and deer).  HEV-specific antibodies were detected in 9.1% of the deer sera and in 1.7% of wild boar serum samples. The present study is the first serological study conducted on red deer in Portugal. The paper is well written and scientifically sound. The paper is well written and goes straight to the essential points of the topic.

However, I would like to ask the authors how they explain a higher seroprevalence value in deer than in wild boar given that the latter are considered the true reservoir of HEV in the environment compared to deer which seem to represent spillovers. 

Minor comments:

In the Introduction the nomenclature should be revised with the use of italics.

In the Material and Methods section, if possible, could the authors provide the sensitivity and specificity of the serological kit used?

Author Response

REVIEWER 3

Q1. Pires H. and coworkers investigated the epidemiology of hepatitis E virus in the Centre of Portugal in sera samples collected from wild ungulates (wild boars and deer). HEV-specific antibodies were detected in 9.1% of the deer sera and in 1.7% of wild boar serum samples. The present study is the first serological study conducted on red deer in Portugal. The paper is well written and scientifically sound. The paper is well written and goes straight to the essential points of the topic.

However, I would like to ask the authors how they explain a higher seroprevalence value in deer than in wild boar given that the latter are considered the true reservoir of HEV in the environment compared to deer which seem to represent spillovers.

A1. The authors thank the reviewer for the interest in our research on the seroprevalence of hepatitis E Virus (HEV) in wild ungulates of Portugal. In our research, we observed a higher seroprevalence of HEV in deer compared to wild boar, despite the conventional understanding that wild boar are often considered the primary reservoirs of HEV in the environment.

A1. Several factors could have contributed to this apparent discrepancy. Multiple genotypes and strains of HEV are known to circulate, and different strains may have varying host preferences. Also, while wild boar are often considered the primary reservoirs in the wild, cross-species transmission can occur from other species. Deer and wild boar may share habitats and water sources, thus facilitating virus transmission between the two species.

To sum up, our study findings warrant further investigation to understand the underlying factors driving the observed seroprevalence patterns in deer and wild boar.

Minor comments:

Q2. In the Introduction the nomenclature should be revised with the use of italics.

A2. Thanks for the comments. The italics have been revised.

Q3. In the Material and Methods section, if possible, could the authors provide the sensitivity and specificity of the serological kit used?

A3. Thanks for the comments. We have added the information of the specificity of the serological kit used.

Round 2

Reviewer 1 Report

Since the study was not performed all over the country, I suggest this title: Prevalence and Risk Factors for Hepatitis E Virus in Wild Boar and Red Deer in northeastern Portugal.

I consider that one question still pending to be answered:

Why you did not take a liver biopsy from the animals that were killed? Why you didn´t try to isolate in culture and/or detect the HEV by molecular biology?

Author Response

Reviewer #1

  1. Since the study was not performed all over the country, I suggest this title: Prevalence and Risk Factors for Hepatitis E Virus in Wild Boar and Red Deer in northeasternPortugal.

Authors’ response (AR): we thank Reviewer 1 for conducting this additional positive evaluation of our manuscript.

Samples were obtained in the Centre region of Portugal. Considering that this is the first detection report of HVE in red deer in Portugal, we would like to maintain the title as: Prevalence and risk factors for hepatitis E virus in wild boar and red deer in Portugal. Alternative: Prevalence and risk factors for hepatitis E virus in wild boar and red deer in the Centre region of Portugal. Anyway, we leave this decision at the Editor’s discretion.

  1. I consider that one question still pending to be answered:

Why you did not take a liver biopsy from the animals that were killed? Why you didn´t try to isolate in culture and/or detect the HEV by molecular biology?

AR: we thank Reviewer 1 for pointing out an interesting issue, but as explained before the aim of this study was to assess seroprevalence in a large number of animals. Taking liver samples was not the focus of the study. Anyway, we will take that valuable suggestion for future investigation. The truth is always the best explanation, and we are sure that the reviewer will understand this circumstance.

Reviewer 2 Report

It appears that Table 1 needs some additional formatting, Hopefully, this will be accomplished before publication.  My concerns have been adequately addressed.

Adequate, a few minor mistakes

Author Response

Reviewer #2

  1. It appears that Table 1 needs some additional formatting, Hopefully, this will be accomplished before publication.  My concerns have been adequately addressed.

Authors’ response (AR): we thank Reviewer 2 for conducting this additional positive evaluation of our manuscript. Table 1 has now been re-formatted.

  1. Comments on the Quality of English Language

Adequate, a few minor mistakes

AR: we have tried to correct those minor mistakes. Thank you!